# Experience of Older Patients with COPD Using Disease Management Apps: A Qualitative Study

**DOI:** 10.3390/healthcare12070802

**Published:** 2024-04-07

**Authors:** Xueqiong Zou, Pingping Sun, Mengjie Chen, Jiang Nan, Jing Gao, Xueying Huang, Yi Hou, Yuyu Jiang

**Affiliations:** Research Office of Chronic Disease Management and Rehabilitation, Wuxi School of Medicine, Jiangnan University, Wuxi 214122, China; 6212807057@stu.jiangnan.edu.cn (X.Z.); 6182806006@stu.jiangnan.edu.cn (P.S.); cmj02u54@rjh.com.cn (M.C.); 6212807061@stu.jiangnan.edu.cn (J.N.); 6212807058@stu.jiangnan.edu.cn (J.G.); 6222807026@stu.jiangnan.edu.cn (X.H.); 6222807101@stu.jiangnan.edu.cn (Y.H.)

**Keywords:** mHealth, older people, chronic obstructive pulmonary disease, qualitative study, chronic disease management

## Abstract

(1) Background: Digital medicine is developing in the management of chronic diseases in older people, but there is still a lack of information on the use of disease management apps in older patients with COPD. This study aims to explore the views and experience of older patients with COPD on disease management apps to provide a basis for the development and promotion of apps for geriatric diseases. (2) Methods: A descriptive qualitative research method was used. Older patients with COPD (N = 32) with experience using disease management apps participated in semi-structured interviews. Thematic analysis was used to analyze the data. (3) Results: Seven themes were defined: (a) feeling curious and worried when facing disease management apps for the first time; (b) actively overcoming barriers to use; (c) gradually becoming independent by continuous online learning; (d) feeling safe in the virtual environment; (e) gradually feeling new value in online interactions; (f) relying on disease management apps under long-term use; (g) expecting disease management apps to meet personalized needs. (4) Conclusions: The adoption and use of disease management apps by older people is a gradual process of acceptance, and they can obtain a wide range of benefits in health and life.

## 1. Introduction

In recent years, respiratory infectious diseases have been prevalent. The immune function of older people is relatively weak, and most of them have chronic diseases, so they are at high risk of respiratory infectious diseases. Chronic obstructive pulmonary disease (COPD) is the most prevalent chronic respiratory disease worldwide [1]. At present, the total number of COPD patients in the world is about 250 million, and the global prevalence is about 10.3% [1,2]. The prevalence increases sharply with age and is highest in people over 60 years old [2]. Therefore, older patients with COPD have a higher demand for non-contact medical care, and the adoption of mHealth can greatly reduce the risk of respiratory disease infection.

The World Health Organization (WHO) defines mHealth as medical and public health practice supported by mobile devices. It covers a variety of services such as mobile health (mHealth), mHealth management, medical information management, and medical decision support [3]. With the rapid development of information and communication technology and the spread of smart electronic devices, mHealth is expected to become a universal reality in the near future. Studies have shown that adoption of mobile information and communication technologies is increasing among older people, and the COVID-19 pandemic has further accelerated the adoption of new technologies among older people, which offers great potential for mHealth to promote the health and well-being of older people [4,5]. In the context of the post-pandemic era, the advantages of mobile health applications (MHA) in the home health management of older people are more obvious. These mHealth interventions based on smartphones and tablets have been gradually applied to the disease management of older patients with chronic diseases [6], which is of great significance in helping older patients to monitor and self-manage diseases, improve medication compliance, promote changes in unhealthy lifestyles, and improve the prognosis of diseases [7,8,9,10].

The application of disease management apps in COPD patients has also achieved good results. Park et al. conducted a study on the effects of a smartphone app-based self-management program on self-care behaviors of COPD patients in Korea, showing that app-based interventions can be used to improve self-care behaviors and physical activity in COPD patients [11]. A meta-analysis study conducted by Chung et al. showed that app-based pulmonary rehabilitation was able to reduce dyspnea symptoms and improve quality of life in COPD patients compared with traditional pulmonary rehabilitation [12]. One such app, myCOPD, was explored several times. The study by Bourne et al. confirmed that the app can improve inhaler use and exercise capacity in severe COPD through an online pulmonary rehabilitation program [13]. Crooks et al. evaluated the effect of the app in patients with mild-to-moderate or recently diagnosed COPD through an RCT study [14].

Although mHealth has many advantages such as acceptability and providing healthcare services to patients beyond time, space, and distance, it provides solutions for older people to cope with the risks of physical aging, chronic diseases, respiratory infections, and other diseases. However, the health inequity caused by the inadequate use of mHealth services for older people caused by the “digital divide” has also attracted the attention of researchers [15]. The barriers and facilitators of mHealth service utilization among older people have been extensively explored [16,17]. Quantitative and qualitative studies have showed that socio-demographic factors (such as age, economic level, etc.), equipment, personal needs, use habits and experience, social support, eHealth literacy, and policies were important factors affecting the adoption and use of mHealth services [18]. By definition, eHealth is the provision or enhancement of health services and information through the Internet and related technologies. Studies have reported that technical training related to the use of mHealth services and the age-appropriate design of mHealth products have positive significance in improving the adoption of mHealth services for older people [19]. In addition, some studies evaluated the usability of mHealth for older people by focusing on user satisfaction, ease of learning, operability, intelligibility, and other indicators [20,21]. To accept and adopt apps for disease management is a kind of behavior change, during which older people face a series of new and unique challenges. However, little is known about the interactive experience between older patients with chronic diseases and disease management apps in disease management.

Therefore, this study aims to explore the experience of older patients with COPD and, using disease management apps, obtain new insights on their participation in this healthcare mode to provide a basis for the development and promotion of older people-specific disease management apps to promote healthy aging. (According to the Chinese division of older people, citizens over 60 years old are older people, so in this study, patients who are over 60 years old are referred to as older patients).

## 2. Methods

### 2.1. Study Design

A descriptive qualitative study was conducted to explore the experiences and perceptions about the use of disease management apps in older patients with COPD. This study followed the consolidated criteria for reporting qualitative studies (COREQ) and was performed in accordance with the Declaration of Helsinki [22] (See Appendix A). Ethical approval for study was obtained from the Medical Ethics Committee of Jiangnan University (JNU20220310IRB17), and this study was supported by the National Natural Science Foundation.

### 2.2. Sample Selection and Recruitment

Purposive sampling was used to select the participants in this study [23]. Participants of different sex, age, and educational level were recruited from two local tertiary general hospitals (affiliated hospital of Jiangnan university and covering the health care services of the entire urban area) from March to October 2023. All participants were recruited directly by the researchers of this study. The inclusion criteria were as follows: (1) age ≥ 60 years old; (2) diagnosed COPD according to the guidelines for the diagnosis and treatment of COPD; (3) experience using disease management apps; (4) ability to communicate in Chinese. Patients with severe dyspnea and severe mental and cognitive impairment were excluded.

The purpose, risks, and benefits of the study and the principles of confidentiality and voluntary participation were explained to the 33 participants either orally or by text. Participants were allowed 1 week to consider whether to participate. One patient declined the interview because of physical reasons, and eventually. 32 participants were interviewed, all of whom signed a consensual consent form.

### 2.3. Data Collection

Prior to the interview, a good relationship was established between the researcher and the participant, and the interview time and place were scheduled after the participant decided to participate in the study. At the beginning of the interview, a self-designed questionnaire designed by the research team was used to collect the general information regarding the participants, including gender, age, education level, and duration of disease management apps use. Based on previous qualitative research (our research team carried out a multi-perspective qualitative study on the perceptions and experiences of telehealth use and online health information use in chronic disease management for older patients with chronic obstructive pulmonary disease in 2021, which provided a certain basis for the development of this study), the interview outline was constructed by the research team and the expert consultation team [24] (See Appendix A). Pre-interviews were conducted with the first 3 participants, which showed that the interview outline did not require further revision, and data from all pilot interviews were included in the final data analysis. To ensure consistency in data collection, all interviews were conducted by the first author of this study (female, with 3 years of learning and practical experience in qualitative research).

Face-to-face semi-structured interviews were used to collect data. All interviews were conducted in quiet places such as the hospital classroom (set up for health care personnel to teach and communicate with medical students), communication room, and the participant’s home, and no third party was involved in the interview process. With the consent of the participants, the complete conversation was recorded with a recording pen to ensure the integrity of the interview material. Emotional and physical changes were also recorded during the interview. The average interview time was 40 min. Data collection and analysis were carried out simultaneously, and after the analysis of 32 interview data, no new themes emerged, the data reached saturation, and data collection was stopped.

### 2.4. Data Analysis

Recordings were transcribed verbatim by the researcher (Z) within 24 h of the end of the interview, and participants were numbered in the interview order (patients: P1, P2...). The transcriptions were checked by carefully listening to the recordings repeatedly, reviewing the transcribed text for accuracy, and sending the text to the respondents for verification. Thematic analysis was used for coding and classification in this study, and qualitative analysis software NVivo 12 Plus was used to manage the data [25]. Two researchers (Z and J) repeatedly read, became familiar with, and immersed in the text materials, extracted meaningful statements, and recorded important statements for primary coding. After the induction and collation of the codes, conceptual themes were formed. If disagreements arose, they were discussed together with the research team until a consensus was reached. Finally, a research team composed of six people reviewed all the themes again, these themes were defined and named, and finally, the study report was generated.

## 3. Results

A total of 32 older patients with COPD were enrolled in this study, including 28 males and 4 females. According to the Global Strategy for the Diagnosis, Management, and Prevention of Chronic Obstructive Pulmonary Disease (2024), the prevalence of COPD in men is significantly higher than that in women, so our sample is more male than female. The basic characteristics of the participants are shown in Table 1. The commonly used disease management apps of COPD patients in this study were “Smart Breath Cloud Health” and “Respiratory Rehabilitation”. These two apps all have similar usage processes and functions, including registration, login, medication management, rehabilitation training knowledge learning, and patient self-reporting. 

Table 2 shows seven themes related to the experience of older patients with COPD using disease management apps extracted from this study, which will be further discussed one by one. We provide examples of how themes emerge; see Table 3.

### 3.1. Theme 1: Feeling Curious and Worried When Facing Disease Management Apps for the First Time 

Most participants described the curiosity and novelty of encountering disease management apps for the first time. Participants reported that positive comments from family members and recommendations from doctors made them curious about the app’s convenience and rich functions. A small number of participants were generally interested in new technologies, and when the disease management apps appeared, it was natural for them to download and use them out of curiosity.


*“My granddaughter always tells me that this app is convenient and easy to use. I am curious to hear her say so.” (P3)*



*“This app was recommended to me by my doctor, he said it has a lot of functions, it can book a doctor’s appointment, buy medicine, and show my body data...... This technology is very new, I’m very curious about it.” (P6)*



*“I’m usually a fan of new technology, and I was definitely curious when this app became available, so I immediately downloaded it to use it.” (P1)*


Participants also expressed concerns about disease management apps. Most of the participants expressed great concern about mHealth apps leaking personal private information, implanting advertisements, and promoting products, and thus falling into fraudulent traps causing property damage. Some participants were also concerned about the quality of telemedicine services provided by the app, and they were worried that the doctors could not physically touch their bodies in the remote context and could not conduct physical examinations for them as if they were face-to-face and realistically feel their pain and moods. Other participants were concerned that technologically advanced health products were complicated to operate and that they would have difficulty learning how to use them. Some participants even mentioned that the learning process may require a lot of time and effort from their family members, and they were worried that it would be a burden to their family members.


*“I’m especially worried about it leaking my privacy, like your phone number. And advertising promotion is also very worried, accidentally encountered fraud to cheat your money.” (P11)*



*“My main concern is whether this app can achieve the effect of seeing a doctor face to face. The doctors across the street could only see me through the screen, couldn’t touch me, couldn’t do a physical examination. They can’t feel some of my pain and emotions.” (P9)*



*“I don’t know how to operate is my biggest worry, now high-tech things are very complicated, and it is a burden to bother my family to teach them again and again.” (P20)*


### 3.2. Theme 2: Actively Overcoming Barriers to Use

Most participants described ways in which they actively overcome barriers to using disease management apps. Some participants made positive preparations for the use of disease management apps and created a good environment for using them, such as purchasing reading glasses and setting up wireless network at home. Some participants reported that careful observation was an effective method, since the use of various types of apps is almost the same, and they could also learn by paying attention to how people around them used them. Some participants also reported that it was the most convenient and quick way to turn to family members when they were in difficulties, and family members were the best teachers. Some participants with extensive smartphone use experience emphasized that they preferred to go online to find out how to use disease management apps correctly on their own, and that they chose to ask for advice only when they had failed many times. Some participants appreciated the regular training courses on the use of smartphones for older people held in the community, where the teaching guidance from professionals and mutual help among peers in the learning group removed many barriers to the use of disease management apps. 


*“My eyesight is bad, in order to see the words on the app, I specially bought reading glasses.” (P2)*



*“I installed a wireless network in my house, (apps) would be easier to use.” (P16)*



*“Look and learn, they are used in similar way, more observation.” (P21)*



*“Ask family is the most convenient and fastest ah, family is your best teacher.” (P30)*



*“I can use a smartphone before, I usually encounter problems first Baidu check, really do not know and then ask others.” (P25)*



*“That training in the community is very useful, it is specially organized for older adults, I often go to listen to it, and when I encounter problems, I ask the professionals, and we usually communicate in the learning group, we can help each other, and a lot of obstacles are solved.” (P18)*


### 3.3. Theme 3: Gradually Becoming Independent by Continuous Online Learning

Participants reported that they learned through the disease management apps and gradually became able to make health decisions independently as they continued to learn. They unexpectedly learned to use a wide range of lifestyle apps that made their usual lives easier and more independent. Most participants mentioned detailed knowledge on disease management apps, including etiology, symptoms, treatment, medication, diet, etc. Their persistence in learning every day facilitated their understanding of the diseases and allowed them to discuss the treatment of the diseases with healthcare professionals without the company of their family members and to choose treatment and care options independently. 


*“There are all kinds of knowledge on this app. I have to learn a little bit every day, why I get this disease, what manifestations, how I should treat it, how to take medicine... Now I know a lot, so I don’t need a child to accompany me to the hospital. I can communicate with the doctor alone, and I can independently choose my own doctor to give me the program.” (P23)*


Some participants learned how to relieve acute asthma attacks on apps and knew how to cope with emergencies, and going out independently became possible. 


*“I used to be afraid to go out alone, I was afraid of having an asthma attack without anyone around, and then I learned how to deal with it on this (apps), and I followed it to practice curled lips breathing, and I would take medication to relieve my asthma with me, and I often go out alone now.” (P8)*


Some other participants emphasized that as they continued to learn that they could use disease management apps proficiently, they unexpectedly found that life apps such as travel, payment, shopping, etc., which used to be cumbersome, became easy to use. They began to deal with bills by themselves, book their own vehicles for traveling, and buy daily necessities by themselves through the apps. They did not need to bother their children with many things, and their life became easier and relaxed, and their happiness increased. 


*“I learned this app, and after using it for a while, I feel that most of the life apps have become simple, and life has become more convenient with these apps. I don’t need my children to buy things for me, I don’t need him to pick me up... I can do it all by myself now, and I feel very happy now.” (P7)*


### 3.4. Theme 4: Feeling Safe in the Virtual Environment

Many participants described feeling safe in the virtual environment when they used the disease management apps. Some participants reported that in the virtual environment everyone had a common identity—apps users—and felt equal to each other without differences in economic levels or social status. 


*“Here we are all apps users, no matter whether you have money or not, what kind of work you used to do, here is equal, everyone will respect each other.” (P17)*


Some participants mentioned that they felt ashamed when discussing the diseases and exposing their pain in front of people they knew well. whereas in the virtual environment of the app, everyone had the same experience of the disease and could empathize with each other more. The atmosphere of conversation was relaxed and pleasant, and they could speak freely. 


*“I usually don’t discuss the disease with my family, friends and neighbors, and I feel embarrassed. Here (in the app), it is different. Everyone has the experience of being sick, no one laughs at you, you can say anything you want, and the overall atmosphere is very relaxed and cheerful.” (P16)*


Some participants emphasized that when communicating with doctors on the app, they could not see the anxious patients waiting outside the clinic or the environment and facilities of the hospital, the inexplicable nervousness brought by the hospital decreased, and they did not have to worry about speaking the wrong words when communicating with doctors. 


*“I prefer to communicate with doctors through this app. I don’t feel nervous or worried about making mistakes. I don’t know why I get nervous every time I come to the hospital to see the patients in line, the equipment and the decorations.” (P24)*


Some participants reported feeling free when using the disease management apps; they could choose the avatars of the app according to their preference, they could freely decide whether to share and post information, and they felt more secure that everything was under their control. 


*“I can choose the avatar of the app and everything, and no one forces me to post anything... I was free. I had everything under control. I felt more secure.” (P9)*


### 3.5. Theme 5: Gradually Feeling New Value in Online Interactions

Participants reported that they used disease management apps to interact frequently and gradually felt their new value during the interaction. Most of the participants said that in the initial stage of using disease management apps, they were “silent” users, who rarely spoke up and chose to silently pay attention to the disease experiences shared by other users on the platform. Through long-term interaction with the app, they gradually realized that they can also share disease information, medication experience, recipes, and exercise management experience to others. When they see that this information can provide help to others, they feel happy, and they gradually realize their own value, have a sense of accomplishment, and feel that they are still a useful person. 


*“In the beginning, I rarely posted my thoughts, I quietly observed the information shared by others, like the young people say “diving”, and I also got a lot of good methods from it. Then slowly I realized that I can also send these messages, it can always help others. Now when I use a new drug or have a new exercise method, I will post it, and I am happy and proud to see them become better.” (P12)*



*“I think even though I’m old and retired, I’m still useful.” (P15)*


Some participants also emphasized that they received help from the interaction, that they had a responsibility to help others, and that they should be a grateful person. Some participants even organize and volunteer regularly. 


*“People should know how to repay kindness, people (on app) helped you, you have to help others ah, this is a kind of responsibility.” (P22)*



*“After using the app, I feel my own value, and now I often participate in community volunteer activities.” (P25)*


Some participants reported that they felt particularly accomplished when their comments were officially adopted and updated in subsequent versions, and felt that they had played their role as an engineer in the development and design of the app. 


*“I once asked a question, and then the staff contacted me, and this question was improved in the later version. I had a great sense of achievement, and I also played the value of an engineer.” (P8)*


Other participants reported that they had made new friends during the interaction, joined some societies, developed new hobbies in new groups, such as painting and singing, and felt they had grown. 


*“I met a new group of friends here (app), I taught them to sing, they taught me to draw... I also grew up.” (P4)*


### 3.6. Theme 6: Relying on Disease Management Apps under Long-Term Use

Participants generally mentioned that disease management apps played the role of health managers to assist them in effective disease management, and they became dependent on them after long-term use. Some participants reported that the app’s medication and exercise reminder function helped them solve the challenge of often forgetting to take medication and exercise and that they could not achieve their current state of health without the help of the app. Some participants emphasized that apps can monitor and record vital signs, which is helpful for tracking their own health trends, seeking timely medical treatment, and accurately describing health problems. Some participants even mentioned that disease management apps have become a necessity for the daily management of the disease, and they keep track of the development of the disease and problems encountered in its management on the app, generating logs on a regular basis, and feeling anxious about the lack of data if their phones are out of power or unable to connect to the internet. 


*“I have a bad memory and often forget to take medicine and exercise, which is no longer the case after using this app. The asthma is not so serious, and my health is getting better. I can’t do without it.” (P7)*



*“If my heart rate is irregular, it can monitor and record it, and if there is a problem, it will alert me, I can go to the hospital in time, and I can directly show the doctor the record.” (P30)*



*“I have been using this app since I was sick. It records my progress of the disease and some obstacles and experiences of disease management. If there is no electricity or Internet, I can’t see this information, and I will be anxious.” (P27)*


### 3.7. Theme 7: Expecting Disease Management Apps to Meet Personalized Needs

Almost all of the participants emphasized that the health apps were developed for the majority of users, and they expressed their expectations for disease management apps from many aspects such as technological breakthroughs, account login, and functional development. Some patients appreciated the app’s voice recognition and control function, saying that it greatly facilitated the use by users with poor eyesight or low literacy levels, but some patients who could not speak Mandarin expressed their expectations for the app’s dialect recognition technology. A portion of the participants reported that the existing apps only support one account login, and they would like to be able to log in to multiple accounts at the same time so that their family members can participate in their disease management together. Participants with multiple diseases also put forward higher requirements for disease-specific management apps, hoping that the respiratory disease management apps can include the management of common health problems in older people, such as diabetes, hypertension, and falls. 


*“Now the technology is very advanced, you can control it (app) by voice, but it can only recognize Mandarin, I can’t speak it, I hope it can recognize dialects as well.” (P6)*



*“This app can only be logged in with one account, and my children cannot see my information. I want them to log in and participate together.” (P13)*



*“I used to use this app because of my asthma, but this year I found high blood pressure again. There are also special apps for the management of hypertension, but I can’t have a problem with the app, now two is OK, more apps easy to mess up in the future, I hope this app also has some common disease management.” (P19)*


## 4. Discussion

This study explored the perception and experience of disease management apps among older patients with COPD. In general, they have an optimistic attitude toward disease management apps, and they actively overcome use barriers through personal efforts and with the help of family, professionals, peers, and other aspects. They also have some personalized expectations for apps. Most patients have a positive experience in the process of using disease management apps and gradually become dependent on them. The study also revealed three new findings: (1) patients gradually became independent through continuous online learning and accidentally acquired the ability to live independently through the use of mHealth apps; (2) the sense of stigma was weakened or even disappeared in the virtual environment, and patients felt more secure; (3) patients gradually felt their own ability in the interaction with apps and actively enjoyed their “new value”.

Previous studies have either focused on the use of digital technology in health or mainly targeted groups of patients with obesity, diabetes, or hypertension, and few studies have focused on patients with COPD [26,27,28]. This study focused on a group of older patients with COPD who could benefit from mHealth and explored their perceptions and experiences of disease management apps. Some studies have extensively discussed the barriers and promoting factors of mHealth use in older adults and proposed that the use of mHealth should be promoted from the aspects of mHealth service design and the improvement of the eHealth literacy of older adults [16,17,19]. From the perspective of benefit perception, this study has identified some new benefits that can promote the continued adoption of disease management apps by older patients.

The use of disease management apps can help patients maintain independence in disease management and daily life. Previous studies have also found that the use of digital health technologies can help patients become more independent in the management of their diseases [18,29]. Building on this, this study found that a large reason why patients were able to maintain their independence was the increased competence that comes with continuous learning and that the disease management apps provided an excellent platform for online learning. This is actually an effective patient activation strategy. Patient activation, which refers to the possession of health-promoting knowledge, skills, and confidence, has emerged as a potential strategy to improve the self-management of patients with chronic diseases [30,31]. Previous studies have found that patients with high activation levels have higher levels of fitness and treatment adherence, and patient activation has emerged as an important predictor of healthcare resource utilization, patient experience, and participation in treatment decisions [32,33,34]. This study adds to this evidence by suggesting that the concept of “skills” in patient activation should be extended to the use of digital health technologies in the context of digital needs. Patients with this skill are more likely to overcome eHealth technical barriers during activation and have an increased likelihood of maintaining independence during disease management. Current patient activation strategies mainly focus on personalized care plans and peer support based on patient activation levels [30]. This study further highlights the potential of the use of mHealth to promote patient activation. The accessibility, availability, and ease of use of mHealth provide more opportunities for patients’ self-management, and the enthusiasm and confidence of patients’ active learning are effectively improved, which is conducive to the successful activation of patients. A study found that patients with high activation levels preferred to use MHA [30]. This suggests that mHealth use and patient activation appear to have a mutually reinforcing relationship. The role of mHealth in patient activation should be fully considered in the future. Further studies are needed to confirm whether existing patient activation strategies are applicable in complex digital environments. In addition, this study newly found the potential of disease management apps’ use in promoting the independent living of patients, but more quantitative studies are still needed to verify the use of MHA and independence.

The study by Raja et al. emphasizes that technology should be easy to use [18]. Studies have pointed to the lack of specialized design guidelines to address perceptual, cognitive, and motor declines in older adults [20,35,36]. To promote the adoption and continued use of MHA by older patients, targeted interface design (large fonts, anthropomorphic icons) and persuasive functional design (reminders, social features, gaming elements, personalized interventions) are progressing [19]. In addition, social support plays an important role in helping older adults overcome barriers to the use of digital technology. Research suggests that professionals such as coaches and health literacy mediators can help patients understand complex health messages in apps [16]. Allowing the younger generation of the family to pass the concept and technology of digital health to older adults and helping the older generation to integrate into the digital society is the embodiment of “cultural regurgitation” in the use of digital technology and new media—“digital regurgitation”. Daily communication and interaction between older peers can inadvertently improve their ability to use mobile technology, which fully reflects the advantages of “intra-generation feeding”. In the process of using digital technology for older adults, it is important to help them overcome obstacles, but the long-term solution is to use the development of artificial intelligence technology to seek technological breakthroughs and reduce the obstacles to technology use by older patients from the source. The Health Information Management and Systems Society’s guidelines for mHealth state that “permissive” systems, that is, systems that can help users avoid mistakes, can facilitate the adoption of mHealth. A systematic review of the role of artificial intelligence (AI) technology in chronic disease management pointed out that data mining, speech recognition, and image recognition by AI technology showed great potential in the diagnosis of diseases [37]. However, most of the current research focuses on healthcare providers (HCPs). In the future, more experience is needed on how to apply the above technologies to older adults to help them achieve the barrier-free use of mHealth and the efficient management of diseases.

Co-creation requires involving all stakeholders in product development and design. Co-creation technologies have been shown to improve the quality, acceptability, and feasibility levels of mHealth products. A scoping review on the theme of co-creation for the health of older people indicated the need for co-creation in specific areas and groups [38]. Research by Leorin et al.’s development team that included people with dementia in the app confirmed that end-user values and life experiences are critical in app development [39]. Previous co-creation experiences in mHealth products have shown that the use of co-creation methods in the design process of health solution products can fully meet user needs and facilitate the wide promotion of products. In recent years, the concept of value co-creation has also evolved from business to the health field [40]. Interaction and resource integration are two key components of value co-creation. Value co-creation allows multiple participants to create value through interaction, and good interaction can help participants become effective co-creators to obtain good service results [40]. This is consistent with the results of the present study. Older patients have frequent interaction and information exchange with other users in the use of disease management apps, which provides conditions for high-quality value co-creation. In addition, older users are willing to put forward their own unique views for the development and design of health products, which is conducive not only to the promotion of products and services but also to helping older users realize their self-worth. The rapid development of mobile technology provides a favorable environment for value co-creation. In an ideal state, digital value co-creation will surpass the limitations of time and space and make interaction and resource integration more convenient, and co-creation efficiency will be effectively improved. However, the corresponding evidence is still lacking. Lundell et al.’s experience of using a participatory co-creation approach to create e-health tools for COPD patients suggests that co-creation in a digital environment is feasible, but value co-creation is not explored further [41]. The deep relationship between value co-creation and self-value realization also needs further exploration. In addition, current research shows that the process of co-creation is different and lacks unified norms and standards [42]. In the future, the process of the older generation participating in co-creation in the digital environment needs to be further explored and standardized, so as to maximize the efficiency of co-creation, generate co-creation value as much as possible, and help them realize their self-worth in the process of co-creation. 

The relationship between digital health technology and patient security has gradually attracted everyone’s attention. Liu et al.’s study found that virtual medical treatment brought insecurity to users [43]. This study found that the insecurity mainly came from the concern of older patients with COPD that remote technology could not replace some face-to-face medical services, resulting in lower medical quality. There are also many studies that find that mHealth can bring people a sense of security [18,44]. A multi-perspective qualitative study by Ekstedt et al. highlighted that patients felt safe and, thus, secure in remote disease monitoring [45]. This study focused on the perceived psychological safety of older patients when communicating with others (including other patients and doctors) in a virtual environment. Common definitions of psychological safety include “feeling able to display and use the self without the fear of negative effects on self-image, status, or career”. A conceptual analysis of psychological safety proposed that psychological safety in the health care work environment influences proactive behavior [46]. Strong interpersonal relationships such as trust, respect, and support can effectively promote psychological safety [46,47]. This could be used to explain why older patients in this study thought that effective interaction in an equal environment created a psychological safety domain for them. The remote environment reduces the chance of face-to-face contact, and patients cannot see the hospital environment and facilities and other patients, which can reduce the tension of patients in the process of medical consultation. In the process of using disease management apps, patients are given the right to make independent choices (such as profile pictures), which can increase their sense of self-control. All of the above are important factors for users to obtain psychological safety. It has been proven that psychological safety can produce positive health care outcomes, and more and more studies have found that psychological safety and age are negatively correlated, so the psychological safety of older patients needs special attention [47]. However, the current research on psychological safety in the field of health care mainly focuses on medical staff and rarely considers patients [48]. Another important reason that this study found for why patients felt psychologically safe was the attenuation or even disappearance of disease stigma, which was due to the opportunity that digital technology brought to reduce barriers related to stigma. This study found that in the MHA designed for specific diseases, the sense of similarity between groups greatly reduced the shame caused by the disease in the process of using it. Batchelor et al. found similar results in a qualitative study on digital psychological education, where users easily gained confidence, hope, and a sense of connection with others in a remote setting, which helped alleviate feelings of isolation and stigma [49]. In the future, the research on psychological safety and stigma in the digital environment should be extended from the population of AIDS patients to the population of older patients with chronic diseases. More research is needed to explore the mechanisms by which digital virtual environments provide psychological safety. While focusing on physical health, the development of mental health mobile products that are suitable for older patients with chronic diseases also needs to be considered.

Although the adoption of mHealth services by older people is on the rise, their use of mHealth services is also influenced by a wide range of personal and external factors. Previous studies have found that the cost of technology is also an important factor, with limited access to broadband or other technology devices for economically disadvantaged populations further hindering access to mHealth services [50]. This is consistent with the findings of this study. These participants in this study were all from a second-tier city, and they tend to have higher education and high economic status. In addition, the local medical service system is of high quality, and as an affiliated hospital of the university, the hospital carries out various cooperative and volunteer service projects with the university, and the social support is strong. This allows participants to have greater access to mHealth services. This is consistent with the findings of Jiang et al. [24]. Therefore, more empirical studies are needed to explore the influence of the profile of older people on the mobile health experience in the future.

The above findings suggest the following: (1) the adoption and use of disease management apps by older people is a process of gradual acceptance. Nursing should identify the stage of patients’ mentality in the process of using disease management apps to promote patients’ self-management and be patient with the changes in patients’ mentality. (2) By using disease management apps, older people not only gain the ability to independently manage themselves and live but also reflect on their social value and improve their expectations of life, which can be used as part of the content of health behavior motivation intervention and the goal of disease management. (3) Some standards should be set in the release and operation of health apps to avoid the concerns of older people in the process of using them. (4) Healthcare providers can encourage patients with stigma to use online communication to communicate with them.

### Strengths and Limitations

This study specifically explored the experiences of older COPD patients using disease management apps through qualitative research. The positive experience and growth of users in the use of mHealth were newly discovered, including the continuous online learning environment provided by the platform, the virtual psychological safety environment, and the realization of self-value brought by interaction. The above findings can provide a broad basis for the development and promotion of disease management apps.

Our study has certain limitations. The sample of this study only includes older patients with COPD with disease management apps use experience, and the specificity of the disease tends to have a certain impact on the experience of mHealth use, so the results of this study cannot represent the experience of the mHealth use of all patients with chronic diseases. Due to the incidence of disease, the gender ratio in this study was not balanced, and opinions from more female participants were missing.

## 5. Conclusions

The adoption and use of disease management apps by older people is a gradual process of acceptance, and they can obtain a wide range of benefits in health and life. In addition to family and social support, technological breakthroughs are the long-term solution to ensure that older people can use mHealth without barriers.

## Figures and Tables

**Table 1 healthcare-12-00802-t001:** Characteristics of participants (*n* = 32).

Characteristics	P ^a^ (*n* = 32)
Gender, *n* (%)	
Male	28 (86)
Female	4 (14)
Age (years), mean (SD)	69.1 (5.6) (61–81)
Education status, *n* (%)	
Primary school and below	15 (47)
Middle school	14 (44)
High school	3 (9)
Higher education and above	0
Smoking habit, *n* (%)	
Yes	25 (78)
No	7 (22)
Disease duration, *n* (%)	
<10 years	21 (66)
≥10 years	11 (34)
Long-term home oxygen therapy (LTOT), *n* (%)	
Yes	5 (16)
No	27 (84)
GOLD grades ^b^, *n* (%)	
GOLD I	0
GOLD II	3 (9)
GOLD III	17 (53)
GOLD IV	12 (38)
The number of exacerbations in the past year, *n* (%)	
0	2 (6)
1	16 (50)
2	10 (31)
≥3	4 (31)
Duration of use of the disease management apps, *n* (%)	
<1 year	18 (56)
≥1 year	13 (44)

^a^ patient. ^b^ GOLD Global Initiative for Chronic Obstructive Lung Disease; GOLD I; FEV1/FVC < 70%, FEV1 ≥ 80%; GOLD II: FEV1/FVC < 70%, 50% ≤ FEV1 < 80%; GOLD III: FEV1/FVC < 70%, 30% ≤ FEV1 < 50%; GOLD IV: FEV1/FVC < 70%, FEV1 < 30%.

**Table 2 healthcare-12-00802-t002:** Seven main themes.

Theme	Description
Theme 1	Feeling curious and worried when facing disease management apps for the first time
Theme 2	Actively overcoming barriers to use
Theme 3	Gradually becoming independent by continuous online learning
Theme 4	Feeling safe in the virtual environment
Theme 5	Gradually feeling new value in online interactions
Theme 6	Relying on disease management apps under long-term use
Theme 7	Expecting disease management apps to meet personalized needs

**Table 3 healthcare-12-00802-t003:** An example of how themes emerge.

Data Extract from Interviews	Codes	Category	Theme
My eyesight is bad, in order to see the words on the app, I specially bought reading glasses. (P2)	Buy reading glasses	Add hardware	Actively overcoming barriers to use
I installed a wireless network in my house, (apps) would be easier to use. (P16)	Installing a wireless network	Add hardware	
Look and learn, they are used in similar way, more observation. (P21)	Observational use method	Self-observed learning	
Ask family is the most convenient and fastest ah, family is your best teacher. (P30)	Seek help from family	Family support	
I can use a smartphone before, I usually encounter problems first Baidu check, really do not know and then ask others. (P25)	Previous experience of using smart phone; Search online independently; Seek help from others	Mobile device use experience; Search online independently; Social support	
That training in the community is very useful, it is specially organized for older adults, I often go to listen to it, and when I encounter problems, I ask the professionals, and we usually communicate in the learning group, we can help each other, and a lot of obstacles are solved. (P18)	Community professional training; Study group communication and discussion	Social support; Peer support	

## Data Availability

Dataset available on request from the authors.

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
