# Peer review of "Experience of Older Patients with COPD Using Disease Management Apps: A Qualitative Study"

_healthcare, 2024, doi:10.3390/healthcare12070802_

Round 1

Reviewer 1 Report

Comments and Suggestions for Authors

Dear Authors,

The topic is very interesting. Hopefully the message will be helpful for patients with COPD, specifically to reduce healthcare use and prevent the progress of an excerbation.

The paper is well written but a bit long. It would have been more balanced including 50% men and 50% women. In my opinion, the paper can be accepted after minor revision. The following suggestions can be considered:

Table 1

It would be better to insert smoking habit and at least GOLD classes or grades. Just confirming that perhaps patients included in the study had mild-moderate COPD. Did you include any patient with LTOT? Did you collect the number of exacerbations per year?

Results

The Results can be described only in a few sentences. A table/box summarising all the themes can be inserted so the paper is more readable.

Author Response

We are pleased by your high affirmation of our work, which will encourage us to undertake further research in this field. We also sincerely appreciate your detailed suggestions on the manuscript, which greatly helped to improve the quality of the manuscript. We have carefully revised each of each comment.

Reviewer 2 Report

Comments and Suggestions for Authors

First of all, I appreciate the opportunity to review the mentioned work.

The topic proposed by the authors is based on a current and increasingly necessary discussion, considering transitions in care, the inclusion of new tools, and obviously, the aging population, which is already a concrete reality in many countries.

Regarding the work, I have only a few minor considerations to highlight, which I leave to the authors' discretion for further justification in the text, namely:

How was consensus reached on the functionalities of the different apps? Did participants use common apps? How can we ensure that the experience was minimally similar? It would be interesting to include more information about the available apps that were used by the participants.

Concerning the data collection procedure: How many participants did the authors approach? Did the hospitals have similar care profiles? Is there any difference between the care systems? Public and private, for example? It would be interesting to provide more information about the hospitals, their care profiles, so that the reader can better understand the type of support the participants received and how the use of the app assisted them.

Regarding the discussion section, the authors present an interesting approach and reallocate current studies. However, little is said about the profile of the elderly and their relationship with the experience. That is, only in the limitations do the authors suggest that the participants' location may have facilitated access. This was an interesting point to consider in the discussion.

Author Response

We sincerely thank you for your constructive comments on our manuscript. These comments were very valuable and we revised the manuscript extensively. Thank you again for your positive comments and valuable suggestions to improve the quality of our manuscript.

Reviewer 3 Report

Comments and Suggestions for Authors

Thank you for the opportunity to review this paper. The results of the interviews are interesting, and the discussion links these themes to larger topics within the field.

MAJOR FEEDBACK:

- A table is needed for the themes to count how many participants mentioned these themes, and how often these themes appeared throughout the interviews.

- There is a disproportionate amount of male participant. These needs to be explained and noted as a limitation.

- It is not clear if all participants are using a specific app known by the researchers, or multiple apps. The details of the apps need to be noted otherwise the themes cannot be applied to all apps for COPD but a specific group of apps, or specific institution app.

MINOR FEEDBACK:

Introduction:

- mhealth is not new or emerging. It has been around for a while and needs to be correctly defined.

- older patient needs to be defined (can be different per country, service, disease).

- introduce and define eHealth.

- What is APP?

- Introduction needs more describing what apps currently exist for COPD - or COPD mHealth specific research.

Methods:

- define local hospitals? (what was the radius or catchment area for this)

- How were the 33 participants screened - were medical teams involved, or did the research team have access or prior knowledge?

- How and why were good relationships established with participants?

- not clear if APP is a specific app?

- Line 102 explain in detail the previous research used to inform this.

- What is a hospital classroom?

- Line 116: should this be no new themes amongst the transcripts?

Discussion

- some acronyms appear that were never introduced (HCP)

Comments on the Quality of English Language

Minor grammar issues spotted throughout.

Some sentence can be re-written to improve flow and communication.

Acronyms need to be consistent in there introduction for example: Chronic Obstructive Pulmonary Disease (COPD) & chronic obstructive pulmonary disease (COPD).

Consistency is needs between APP, APPs, apps.

Use mHealth consistently throughout paper instead of mobile health.

Cited paper authors need to be introduced correctly. Raja M's should be Raja et al.

Author Response

We sincerely appreciate your comments, which will help to improve the quality of our paper. We also appreciate the opportunity to further clarify the data collection and analysis methods used in this study. This misunderstanding of the qualitative methodology may have occurred because we did not explain the methodology sufficiently and in detail. This study followed the consolidated criteria for reporting qualitative studies(COREQ), and the COREQ in Appendix 2.

Round 2

Reviewer 3 Report

Comments and Suggestions for Authors

Thank you to the authors for their detailed response and for the work put into answering my comments.

I appreciate the authors would prefer not to prefer the table of themes and occurrences. In this case, could you add your coding tree as a diagram.

Lastly could the details provided in Answer 2, 5, 13, 14 be added to the manuscript. Not all readers will be familiar with this and would be excellent for highlighting context for anyone interested in using your research in future.

Comments on the Quality of English Language

minor editing needed

Author Response

(The authors gave the same response as above.)
